# Effect of Soil Acidification on the Production of Se-Rich Tea

**DOI:** 10.3390/plants12152882

**Published:** 2023-08-07

**Authors:** Bin Yang, Huan Zhang, Wenpei Ke, Jie Jiang, Yao Xiao, Jingjing Tian, Xujun Zhu, Lianggang Zong, Wanping Fang

**Affiliations:** College of Horticulture, Nanjing Agricultural University, Nanjing 210095, China2021104080@njau.edu.cn (H.Z.); kwp1464421789@126.com (W.K.); 2019201009@njau.edu.cn (J.J.); zi_su1@163.com (Y.X.); tianjingjing@njau.edu.cn (J.T.); zhuxujun@njau.edu.cn (X.Z.); zonglg@njau.edu.cn (L.Z.)

**Keywords:** tea plantation soil, acid rain, acidification, Al, Se

## Abstract

Selenium (Se)-enriched tea is a well-regarded natural beverage that is often consumed for its Se supplementation benefits. However, the production of this tea, particularly in Se-abundant tea plantations, is challenging due to soil acidification. Therefore, this study aimed to investigate the effects of changes in Se under acidified soil conditions. Eight tea plantation soil monitoring sites in Southern Jiangsu were first selected. Simulated acid rain experiments and experiments with different acidification methods were designed and soil pH, as well as various Al-ion and Se-ion concentrations were systematically determined. The data were analyzed using R statistical software, and a correlation analysis was carried out. The results indicated that as the pH value dropped, exchangeable selenium (Exc-Se) and residual selenium (Res-Se) were transformed into acid-soluble selenium (Fmo-Se) and manganese oxide selenium (Om-Se). As the pH increased, exchange state aluminum (Alex) and water-soluble aluminum (Alw) decreased, Fmo-Se and Om-Se declined, and Exc-Se and Res-Se increased, a phenomenon attributed to the weakened substitution of Se ions by Al ions. In the simulated acid rain experiment, P1 compared to the control (CK), the pH value of the YJW tea plantation decreased by 0.13, Exc-Se decreased by 4 ug mg^−1^, Res-Se decreased by 54.65 ug kg^−1^, Fmo-Se increased by 2.78 ug mg^−1^, and Om-Se increased by 5.94 ug mg^−1^ while Alex increased by 28.53 mg kg^−1^. The decrease in pH led to an increase in the content of Alex and Alw, which further resulted in the conversion of Exc-Se to Fmo-Se and Om-Se. In various acidification experiments, compared with CK, the pH value of T6 decreased by 0.23, Exc-Se content decreased by 8.35 ug kg^−1^, Res-Se content decreased by 40.62 ug kg^−1^, and Fmo-Se content increased by 15.52 ug kg^−1^ while Alex increased by 33.67 mg kg^−1^, Alw increased by 1.7 mg kg^−1^, and Alh decreased by 573.89 mg kg^−1^. Acidification can trigger the conversion of Exc-Se to Fmo-Se and Om-Se, while the content of available Se may decrease due to the complexation interplay between Alex and Exc-Se. This study provides a theoretical basis for solving the problem of Se-enriched in tea caused by soil acidification.

## 1. Introduction

Selenium (Se) is essential for human health, playing a significant role in the body’s detoxification processes. It helps remove free radicals, remove contaminants from waste, and inhibits the creation of peroxide esters [1]. A lack of Se can cause health problems such as Keshan disease (KD) and white muscle disease [2]. The bioaccumulation of Se in plants is critical for Se supplementation and the protection of human health [3,4]. Se-rich foods such as rice, flour, edible fungi and algae, meat, and tea serve as essential sources of this element since the human body is incapable of producing it [5,6]. Currently, Se-enriched tea has been successfully produced in Shaanxi and Guizhou tea estates, and China has set a content criterion for such tea [7]. However, despite being Se-rich, the southern region of Jiangsu province has been unable to produce Se-enriched tea [8]. Many factors influence plant Se content, including plant physiology, plant species, soil Se content, speciation, and bioavailability [9]. Soil, being a significant repository of Se, serves as a critical material basis for Se to enter the food chain through plants [10]. The total Se concentration and its form in the soil are contingent on the soil solution’s pH and the parent material [11]. Se in the soil mainly exists as water soluble selenium (Sol-Se), exchangeable selenium (Exc-Se), acid-soluble selenium (Fmo-Se), manganese oxide selenium (Om-Se), and residual selenium (Res-Se) [12]. Exc-Se and Sol-Se are the most effective for plant uptake of these forms, and their combined amount is regarded as the accessible Se in the soil [13]. Meanwhile, Se is present in diverse forms, particularly as selenide [Se (-II)], elemental selenium [Se (0)], selenite [Se (IV)], or selenate [Se (VI)], all of which exist naturally [14]. Changes in pH have a major impact on selenite adsorption in soil, principally via altering the valence state of heavy metals [15]. According to the current study, Se can limit heavy metal uptake via immobilizing the compounds generated by plants. Se antagonism with heavy metals (As, Cd Cr, Hg, Sb, and Pb) in the soil is connected to the soil pH [16]. Therefore, the soil’s pH value is a key determinant of the bioavailability of Se in soil [17].

Soil acidification in recent years has led to a significant drop in soil pH, resulting in a multitude of soil-related issues [18]. Notably, tea plantations have experienced the highest rate of soil acidification, with a marked decrease in pH [19]. Furthermore, as the tea plant cultivation time lengthens, soil acidity rises, providing challenging conditions for tea plant growth [20]. Soil acidification in tea plantations can be ascribed to a variety of factors including acid rain, the return of fallen and pruned leaves to the field, the secretion of organic acids, and regular fertilization practices in tea plantation management [21]. Studies have found that tea plants grow considerably slowly under the impact of acid rain with a pH of 3.0, especially when combined with a moderate acidity of 30 mg L^−1^ of aluminum (Al) addition [22]. Additionally, selenite adsorption to soil is high but decreases as the pH increases [23].

According to research on the distribution coefficient of Se in Japanese agricultural soils, active Al is the primary adsorbent for Se [24]. Some studies have revealed that organic matter and mineral Fe/Al oxides play an essential role in conservation of soil Se [25]. Tea plants are typical acidophilic and Al tolerant plants, and Al required for tea plant growth is considered a beneficial mineral [26]. The current study classifies Al in soil into five forms: water soluble aluminum (Alw), exchange state aluminum (Alex), organic bound aluminum (Alo), inorganic state aluminum (Alino), hydroxide aluminum (Aloh), and humic acid aluminum (Alh) [27]. The active action of Al is as follows in the order from strong to weak: Al^3+^ > Al (OH)^2+^ > Al (OH)_2_^+^ > Al (OH)_4_^−^ [28]. When exploring the influence of soil pH on Al absorption by tea plants, an increase in soil pH from 3 to 6 leads to a decrease in the extractable Al concentration in the soil [29]. The solubility of Al^3+^ significantly increases in the acidic soil of tea gardens. Al^3+^, being the most abundant form, profoundly impacts plant growth and readily binds to organic matter [30]. There have been studies using an Al–Se co-precipitation system to remove selenite ions from water. The principle is that the compound Al_2_ (SeO_3_)_3_ formed by Al^3+^ adsorption of the selenite negative ion SeO_3_^2−^ is insoluble [31]. However, the factors influencing the transformation of Se morphology in tea plantation soil, as well as the underlying mechanisms of this transformation remain largely unknown. Therefore, this paper speculates that in the case of soil acidification in tea gardens, some chemical changes occur between Se and Al in the Se-rich tea plantations, resulting in the inability of tea plants to absorb Se in the soil.

In order to explore the effect of soil acidification on the production of Se-rich tea, we designed experiments using field sampling to simulate acid rain and under various acidification conditions. The specific objectives of the study include the following: (1) To explore the morphological transformation of Se and Al under varying soil pH conditions in tea plantations under natural circumstances; (2) to investigate the morphological transformation of Al and its effect on the bioavailability of Se in tea plantation soil during pH changes simulated by acid rain; (3) to analyze the relationship between the morphological transformation of Al in tea plantation soils under various acidification methods and bioavailability of Se. This research contributes to our understanding of the relationship between Se and Al morphology with respect to soil pH in tea gardens. It also provides a theoretical foundation for producing Se-rich tea in Se-rich tea plantations, which will help to address soil acidification challenges.

## 2. Results and Discussion

### 2.1. Effect of Variation in Soil pH of Tea Plantations with Different Forms of Se

Research has demonstrated that certain crops, including cabbage, wheat, and spring rape, can accumulate Se in Se-rich environments, thus becoming Se-enriched crops [32,33]. Similarly, tea plants can yield Se-rich tea when grown in Se-rich soils. However, in our study area, despite the Se-rich soil, the produced tea was not Se-rich (Appendix A). To explore the reasons affecting the production of Se-rich tea, we determined the pH value and Se content in soil and Se content in tea from eight samples of tea plantations over three years (Figure 1).

The results show the morphological distribution of Se, as follows: Res-Se > Om-Se > Fmo-Se > Exc-Se > Sol-Se. We can see that in Yangxian Tea Farm (YX), Furong Tea Farm, Gaomiao Tea Farm (GM), Tea Science Institute (CKS) of Om-Se > Fmo-Se > Exc-Se. However, in Yan Jingwan Tea Farm (YJW), Shengdao Tea Farm (SD), and Lingxia Tea Farm (LX), the proportion of Exc-Se is greater than that of Om-se and Fmo-Se. The proportion of Sol-Se in YX and GM was also found to be increased. Exc-Se and Sol-Se were the main components of available Se, and the proportion of available Se increased with the increase of pH. The contents of Om-se and Fmo-Se decreased with the decrease of pH (Figure 1a). Correlation analysis revealed a positive correlation between Exc-Se and Res-Se and a significant positive correlation between Fmo-Se and Om-Se. Negative correlations were found between Exc-Se, Fmo-Se, and Om-Se, as well as between Res-Se, Fmo-Se, and Om-Se. Furthermore, Res-Se was negatively correlated with Om-Se (Figure 1b). The pH showed a positive correlation with Exc-Se and Res-Se (Figure 1c).

According to the result analysis in Figure 1, Res-Se is the main form of Se in tea plantations. It has been shown that the parent material of soil is the key factor in determining the form of Se in soil, and the parent material of soil is mostly insoluble Res-Se [17]. In addition, when the pH in the soil is reduced, the Se form will be easily absorbed by plants into a form that is difficult for plants to absorb [34]. Upon integrating these findings with our results, we observed that when the pH decreased, Exc-Se and Res-Se morphed into Fmo-Se and Om-Se. Conversely, as the pH increased, Fmo-Se and Om-Se transformed into Exc-Se and Res-Se. These outcomes align with previous studies [35], further indicating that soil pH impacts the transformation of soil Se forms in tea plantations. Depending on the strength of correlation, Fmo-Se and Om-Se may be better transformed to Res-Se than Exc-Se as effective Se.

### 2.2. Effect of Variation in Soil pH of Tea Plantations on Different Forms of Al

The tea plant is a typical acidophilic plant, and Al element plays a crucial role in its growth, secondary metabolism, and material exchange [26]. To understand if the form of Al influences the transformation of Se, we assessed various forms of Al in the soil (Figure 2a). The morphological distribution of Al was basically the same, Alh > Aloh > Alex > Alo > Alw > Alino. The high proportion of Alh and Aloh content correlates to the fact that the eight sampled tea farms are located in the mountainous areas of Yili and the Ningzhen Mountains, which correlate with the shale slope parent material [36].

Our results (Figure 2b) demonstrated a negative correlation between Exc-Se, Alex, and Alw. There was a significant positive correlation between Fmo-Se and Aloh. There was a significant positive correlation between Fmo-Se and Alw. Om-Se was positively correlated with Alw. Fmo-Se and Om-Se were positively correlated with Alex. The Res-Se was negatively correlated with Alw. Res-Se was negatively correlated with Alex. The Res-Se was negatively correlated with Alo. The results show that pH is positively correlated with Exc-Se and Res-Se. The pH was negatively correlated with Fmo-Se and Om-Se. The pH was negatively correlated with Alex and Alw. Exc-Se and Res-Se were negatively correlated with Alex and Alw. The Fmo-Se and Om-Se were positively correlated with Alex and Alw. But the correlation between Res-Se and pH was stronger than for Exc-Se and pH. The correlation between Res-Se and Om-Se was stronger than that between Res-Se and Fmo-Se (Figure 2c).

The pH value is an important factor affecting the existence of Al morphology. When the soil value is pH < 4, the fixed Al in the soil is activated, and the active Al mainly exists in the form of Al^3+^. Alex is mainly composed of Al^3+^, which is easy to combine with organic matter to form Alo and inorganic matter to form Alino [20]. Similarly, the Alw also has a significant negative correlation with Exc-Se and Res-Se, which also lead to a decrease in Exc-Se and Res-Se. However, when the pH increased, Alex and Alw decreased, Fmo-Se and Om-Se decreased, Exc-Se and Res-Se increased, because the substitution effect of Al^3+^ on the Se ion is relieved [37]. Therefore, we can speculate that increasing the pH value of acidic soil can increase the content of Exc-Se in the soil, and the tea tree can absorb and accumulate more Se element. Similarly, even in acidic soils, Exc-Se absorption by tea plants can be alleviated by removing Al^3+^ from the soil. Both methods are beneficial to the production of Se-rich tea.

### 2.3. Effect of pH Change on Se Morphology under Simulated Acid Rain

Through field investigation results, we found that pH may be related to Se morphology transformation in soil. Acid rain plays an important role in soil acidification and is the main cause of soil pH reduction in tea plantations [19]. With the continuous development of industrialization in southern China, acid rain has had an increasingly important effect on soil pH reduction in tea plantation in southern Jiangsu province [38]. To test these conjectures, we designed an acid rain experiment.

Our results revealed a consistent trend in the Exc-Se content across four soil samples, when the pH was lower than the control (CK); the Exc-Se content was also lower. As the pH increased, the Exc-Se content gradually rose. Fmo-Se content in four soil samples saw a significant increase only under the P4 treatment. Om-Se content increased with decreasing pH. The trend in Sol-Se content in the YX and CKS soil samples was inversely related to pH, with Sol-Se content in YX samples decreasing as pH increased. Res-Se content in four of the soil samples declined as the pH rose (Figure 3a). The results showed that Exc-Se was positively correlated with Res-Se in soil samples, but only YX and SD were significant. Meanwhile, there was a significant positive correlation between Fom-Se and Om-Se in YJW and YX soil samples. Only YJW soil sample was consistent with the results of the tea plantation experiment; Exc-Se showed a significant negative correlation with Fmo-Se and Om-Se (Figure 3b).

These results indicate that, among the four soil samples, only pH and Res-Se exhibit a positive correlation, while pH and Om-Se display a negative correlation. This suggests that the drop in pH caused by acid rain is only one of the reasons for the mutual conversion between the various forms of Se. In comparison with the results in Section 2.1, the data shows that pH is positively correlated with Res-Se and negatively correlated with Om-Se in the transformation of Se forms in tea plantation soil (Figure 3c). This is due to the fact that Se in soil is mostly in the form of selenite. This can convert to Se^2−^ when soil acidity is low [39]. The above results show that the decrease of total Se content is mainly due to the decrease of Exc-Se and Res-Se, and Fmo-Se and Om-Se. Simultaneously, the conversion of Exc-Se to Fmo-Se and Om-Se was significant [20]. The decrease of pH caused by acid rain alone could not directly reduce the decrease of Exc-Se in soil, and the effect on Se absorption by the tea plant was not significant.

### 2.4. Effect of pH Change on Al Morphology under Simulated Acid Rain

The acid rain treatment at varying pH levels showed that the concentrations of different Al forms in the YJW tea plantation soil were higher than in CK. Alh levels decrease as pH increases, as do Alex levels. When the pH decreased, the Aloh content in the YX tea plantation increased, while the opposite was true for the Alex content. The Alex contents in P2–P4 were lower than in CK. The P1 and P4 in the SD tea plantation exhibited lower Alh levels than CK, whereas P2 and P3 showed the opposite. Aloh content initially decreased and then increased, with only the P4 showing higher levels than CK. Alex content declined as the acidity of the acid rain increased. The Aloh content in all tea plantations was lower than in CK, and the Aloh content increased in P1–P3, but decreased in P4. The Alion and Aloh content increased under P4. The Alex content decreased with increasing pH, falling below CK levels. Under P4, the Alion and Aloh content increased. The Alex content decreased with decreasing pH, falling below CK levels (Figure 4a).

In the CKS soil samples, Alo exhibited a significant negative correlation with Res-Se. Aloh was significantly positively correlated with Fmo-Se. Alex had a significant negative correlation with Sol-Se. In the SD soil samples, Alw was significantly positively correlated with Om-Se. Alex showed a significant negative correlation with Om-Se. In the YJW soil samples, there was a notable negative correlation between Alw and Exc-Se. Alh was negatively correlated with Exc-Se and Sol-Se, but positively correlated with Om-Se. In the YX soil sample, Alo showed a significant positive correlation with Fmo-Se, Om-Se, and Sol-Se. Alex was significantly negatively correlated with Exc-Se and Om-Se, but positively correlated with Sol-Se (Figure 4b). Our results showed that when the pH decreased, the Aloh and Alh levels declined, while the Alex content increased. A significant negative correlation was observed between Aloh, Alh, and Alex (Figure 4c). This may be due to the increase of H^+^ content caused by acid rain, which leads to the fracture of the inter-layer Si-Al bond, the formation of Alh and Aloh, and the hydrolysis of Alh and Aloh into Alex under H^+^. As the acidity of acid rain increases, H^+^ in acid rain interacts with interlayer Al to form Alino, and Alino reacts with H^+^ to form more exchangeable Al [40]. Aloh was shown to be adversely linked with Fmo-Se and Om-Se. On one hand, Aloh has a higher density of active sites and has a greater adsorption effect on Om-Se, and Om-Se has a stronger chemical affinity for Aloh. On the other hand, Aloh forms complexes with Om-Se through ligand exchange, and OH^−^ is produced at the same time, resulting in the dissolution of Aloh and the release of Al^3+^ to produce a large amount of Alex [41]. It was also found that Alex and Alw were significantly negatively correlated with Exc-Se and Sol-Se. This may be due to the complexation of Al^3+^ in Alex and Alw and SeO_3_^2−^ in Exc-Se and Sol-Se [42].

### 2.5. Effect of pH Change on Se Morphology under Different Acidification Modes

The pH lowering trend is produced by several acidification modes, as well as the changes and correlation of Se and Al. The results agree with those of the field test and acid rain test. Under various acidification treatments, soil pH exhibited the most significant decrease under T6 (Appendix A). Compared with CK, T1 reduced the content of Sol-Se and Fmo-Se, and increased the content of Exc-Se and Om-Se. T2 and T3 reduced the content of Sol-Se, and increased the content of all other forms of Se. T4 reduced the content of Res-Se, Sol-Se, and Exc-Se, and increased the content of Fmo-Se and Om-Se. Om-Se increased under T6, while all other forms of Se decreased (Figure 5a). Fmo-Se exhibited a positive correlation with Om-Se, Exc-Se showed a positive correlation with Res-Se, while Om-Se was negatively correlated with both Exc-Se and Res-Se (Figure 5b). The pH was negatively correlated with Fmo-Se and Om-Se, but positively correlated with Exc-Se (Figure 5c).

Throughout the T1–T6 process, the pH consistently decreased, leading to a decrease in total Se content. Exc-Se content also continuously decreased, and Exc-Se was positively correlated with pH. Despite the decline in total Se content, the content of Om-Se and Fmo-Se still increased. Therefore, it can be inferred that Om-Se and Fmo-Se are converted from another Se content. In line with the correlation analysis results, Exc-Se and Res-Se can transform into Om-Se and Fmo-Se when the pH decreases. Sol-Se content was lowest under T4 and T6, indicating that a decrease in pH could lead to a reduction in Sol-Se content, with acid rain being a major contributor to this decline. However, Sol-Se significantly increased under T2, suggesting that dead leaves foster the production of Sol-Se. 

### 2.6. Effect of pH Change on Al Morphology under Different Acidification Modes

T2 decreased the content of Alh and increased the content of Alo, Alex, and Alw. The different Al forms increased under T3, the content of Alh, Alo, Alex, and Alino increased under T4. The content of Alh, Alo, Alex, Alw, and Alino increased under T5 and T6, and the contents of Alh, Alex, and Alw were higher and the contents of Alo and Alo were lower under T6 compared to T5 (Figure 6a).

Under different acidification treatments, Alw was highly significantly and positively correlated with Om-Se, and Alo was significantly and positively correlated with Fmo-Se, Om-Se, and Sol-Se. Alion was significantly and positively correlated with Fmo-Se and Om-Se, and so was Alex. The Alh was significantly and positively correlated with Om-Se and negatively correlated with Res-Se (Figure 6b). The pH was positively correlated with Exc-Se and Alh with Aloh. The pH was negatively correlated with Om-Se and Alex, and the pH was negatively correlated with Fmo-Se and Alino (Figure 6c).

In T1, NH_4_^+^ was nitrated to form NO_3_^−^. At the same time, base ions leaching from the soil colloid surface, result in a soil colloid surface charge imbalance and more and more H^+^ is adsorbed on the soil colloidal surface. Some Al-ions have difficulty in combining with organic acids and bring about a decrease of Alo content [43]. The content of active Al in the soil of the tea plantation can be reduced by the return of tea plant wilt to the soil under T2, and the content of tea polyphenol (TP) in tea plant wilt can reach 15–20%. TP have strong antioxidant effects, which can eliminate free radicals and chelate Al^+^, resulting in the increase of Alo and Alion content. It is worth noting that the contents of Om-Se and Fmo-Se increased, and there was a significant positive correlation with Alo and Alion. In addition, Alw and Sol-Se were elevated at the same time, which may be closely related to the decomposition of biological organic matter in the tea plant wilt [44]. Reduction of Alion content in T3 and organic acids in root secretions is through the action of protons and organic acid ions. Various Alino in soil colloids in a colloidal state are activated by H^+^ and converted to Alex by ion exchange. Citric acid increases the content of Alex by adsorbing anions to the variable charge soil surface, thereby reducing the positive charge of the soil surface [45,46]. The H^+^ in acid rain is adsorbed on the surface of soil particles through a base ion exchange reaction with the soil colloidal surface, and then spontaneously reacts with the Al on the mineral lattice surface, causing soil acidification and rapidly transforming into Alex [47].

According to the results, T2 has a significant impact on the development of Alo, and T4 can effectively degrade the production of Alo. This is because H^+^ in acid rain has the effect of dissolving Alo or hindering its formation. T2 and T3 played an important role in the formation of Alw. In addition, compared with T4 and T6, acid rain was the main factor affecting Alion content, and there was a significant positive correlation between Alion and Om-Se. T6 treated with the lowest pH value, Alion decreased but did not affect the increase of Om-Se. Upon conducting a correlation analysis, we found a significant negative correlation between Exc-Se and Om-Se, while Alex showed the opposite trend. The decline in Se concentration in the soil solution may be due to enhanced adsorption of Se from soil solid; the lesser Se is oxidized and adsorbed in the soil in the form of Om-Se [3]. Concurrently, it was found that the adsorption modes of selenite and Al oxide differ under strong acid conditions [48,49]. In acidic soil, Al^3+^ is released from the solid phase of the soil. The compound Al_2_ (SeO_3_)_3_ formed by Al^3+^ adsorbing the negative selenite ion SeO_3_^2−^ is insoluble. These results indicate that it is difficult for acidified Se-rich soil to provide Se that can be absorbed and utilized by tea plants, mainly due to the increase in Al^3+^ caused by acidification and the complexation of SeO_3_^2−^ in Se-rich soil. T6 is a complex acidification method that is closer to actual production, while trying to solve the harm caused by T6 can be the key to solving the problem.

## 3. Materials and Methods

### 3.1. Study Area and Sampling

Jiangsu province, a crucial part of the Yangtze River Delta, is situated in the heart of the eastern coastal region of China (30°45′–35°20′ N, 116°18′–121°57′ E). The region boasts a typical subtropical monsoon climate and moderate rainfall, with an average annual temperature ranging between 13 and 16 °C. The tea plantation area covers approximately 34,100 hectares [39].

In this study, tea plantation soils were selected from eight tea plantation soil monitoring sites in Jiangsu Province for three years (2012, 2015, and 2018). The sampling points were distributed among SD; YX; FR; CKS; LX; Linggu Tea Farm (LG); GM and YJW (Figure 7a and Appendix A). A five-point sampling method was used to collect 0–20 cm topsoil samples. The resulting soil samples were stored in sealed bags for component analysis.

### 3.2. Design of Acid Rain Treatment with Different pH

Soils for simulated acid rain trials were taken from SD, YX, CKS, and YJW tea plantations in 2018. Based on the main components of acid rain in Jiangsu, the electrolyte master batch was prepared as outlined below according to the following: (n(NaF):n(CaSO_4_):n(NaCl):n(NH_4_Cl):n(NH_4_NO_3_):n(KCl):n(MgSO_4_) = 2:23.5:5:6:8.5:2:2) [50]. Deionized water was used as CK and four soil samples were treated with four simulated acid rain samples (P1: pH = 4.09, P2: pH = 4.51, P3: pH = 4.98, P4: pH = 5.33) solutions. The simulated acid rain solution was added to 5 mL every three days and repeated 3 times each time. Each treatment contained three biological replicates. The simulation test lasted for 6 months, and the soil samples were removed and dried naturally (Figure 7b).

### 3.3. Design of Treatment with Different Acidification Methods

The test soils for simulating different acidification methods were taken from YJW in 2018. Six treatments were set up: CK, T1: 0.318 g of urea was applied according to the average annual fertilizer application in tea plantations. T2: 3.07 g of tea plant pruning material was applied according to the average annual pruning volume of tea plants in tea plantations. T3: tea plant root secretions mainly consisting of an application rate of 0.02 g of succinic acid and 0.01 g of citric acid. T4: The simulated acid rain solution was prepared as in P4. T5: combined T1, T2, and T3 treatments. T6: combined T1, T2, T3, and T4 treatments (Figure 7c). Each treatment contained three biological replicates. The simulation test lasted for 6 months, and the soil samples were taken out and dried naturally.

### 3.4. Determination of Soil Samples

Soil pH was determined by the potentiometric method (water–soil ratio 2.5:1) [51]. A continuous gradient extraction was used to extract different forms of Al and total Al content according to Barão et al. [52]. In brief, (1) Alex: weigh 1 g of air-dried soil sample in a 50 mL plastic centrifuge tube, add 10 mL 1 mol L^−1^ KCl. (2) Alino: add 10 mL 0.2 mol L^−1^ HCl solution; (3) Alo: add 20 mL 0.1 mol L^−1^ Na_4_P_2_O_7_ solution. (4) Aloh: add 10 mL of 0.5 mol L NHAc solution to the Alex extraction residue, fix the volume with 0.02 mol L^−1^ HCl. (5) Alh: add 10 mL 0.2 mol L^−1^ NaOH solution to the Aloh extraction residue, and fix the volume with 0.2 mol L^−1^ HCl. (6) Alw: according to the soil and water ratio of 5:1 extraction. The sample solutions were subjected to inductively coupled plasma-mass spectrometry (ICP-MS) with a machine to determine measurements [46].

Total Se: 0.5 g of soil sample was weighed in a polytetrafluoroethylene crucible, digested with HNO_3_-HClO_4_-HF, and measured using an atomic fluorescence photometer [53]. Different forms of Se were extracted by a continuous graded extraction method. (1) Sol-Se: weigh 1 g of air-dried soil sample in 20 mL 0.25 mol L^−1^ KCl. (2) Exc-Se: 20 mL 0.7 mol L^−1^ KH_2_PO_4_. (3) Fmo-Se: 20 mL of 2.5 mol L^−1^ HCl. (4) Om-Se: Add 8 mL of 5% K_2_S_2_O_8_ and (1:1) HNO_3_ mL. (5) Res-Se: HNO_3_^+^, HClO_4_^−^ mixed acid (4:1) 160–170 °C digestion until soil is grayish white [54].

### 3.5. Data Processing and Statistical Analysis

Microsoft Excel 2016 [55] and R version 2.2.1 [56] were used to statistically analyze the contents of Se and Al in tea and soil in different forms, with the changing data of soil pH values. Origin 8.5 (OriginLab, Northampton, MA, USA) software was used for the analysis of variances (ANOVA) for Se and pH changes in tea and soil [57]. Pearson correlation coefficient was calculated to better understand the correlation between pH and Se in different soil forms, the pH directly with different Al forms, and the different Se forms with different Al forms in acidified soil [58]. Arc-Map (Environment System Research Institute, Redlands, CA, USA) was used to process the coordinate data of sampling points and obtain the schematic diagram of sampling points [59].

## 4. Conclusions

In the tea plantation soil examined in this study, Se predominantly exists as Res-Se, while Al mainly takes the form of Alh and Aloh. When the pH value decreases, the total Se content continually drops, with Res-Se and Exc-Se being the primary contents. When pH drop induced by acid rain was simulated, Exc-Se and Sol-Se, which plants readily absorb, were transformed into Fmo-Se and Om-Se, which are harder for plants to absorb. This leads to the conversion of Aloh and Alh to Alex. Furthermore, soil acidification can result in complexation effects of Alex and Exc-Se, thereby affecting tea plant absorption of Se from the soil.

Our study is the first to analyze the relationship between different forms of Se and Al in acidified soil in tea plantations. It offers a theoretical foundation for addressing the challenge of Se enrichment in tea brought about by soil acidification. It also presents a novel approach to producing high-quality, safe, Se-rich tea in the Jiangsu province of China. Additionally, more research is still needed to elucidate why Se-rich soils are incapable of producing Se-rich tea.

## Figures and Tables

**Figure 1 plants-12-02882-f001:**
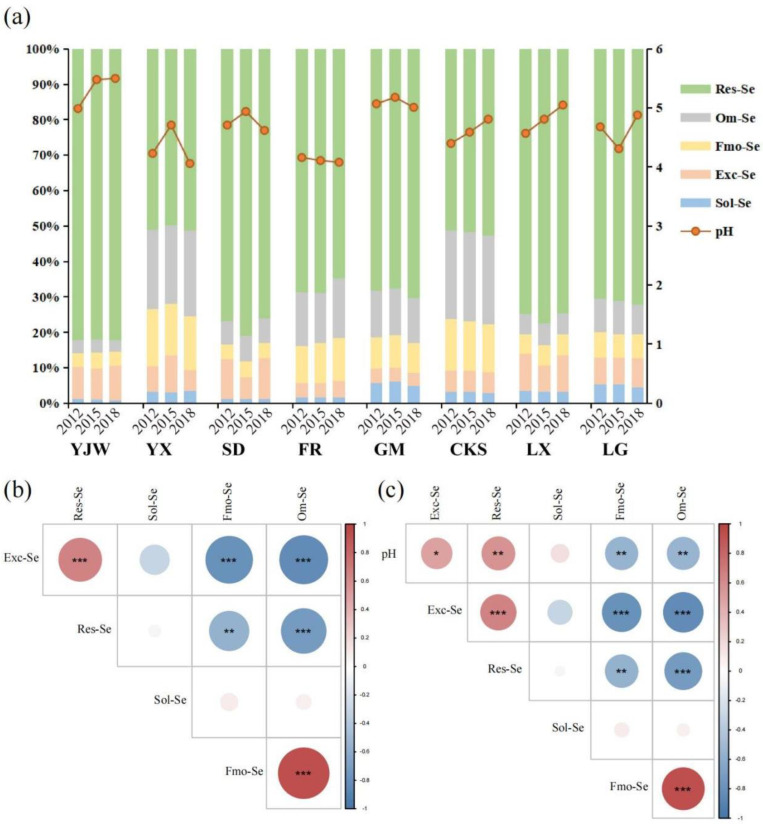
An illustration of the impact of soil pH fluctuations on Se morphology in tea plantations: (**a**) The relationship between Se morphology and pH values across eight sampling sites over a three-year period; (**b**) the correlation between different forms of Se; (**c**) the correlation between pH levels and various forms of Se. The symbols *, ** and *** indicate statistical significance at *p* < 0.05, *p* < 0.001 and *p* < 0.0001, respectively.

**Figure 2 plants-12-02882-f002:**
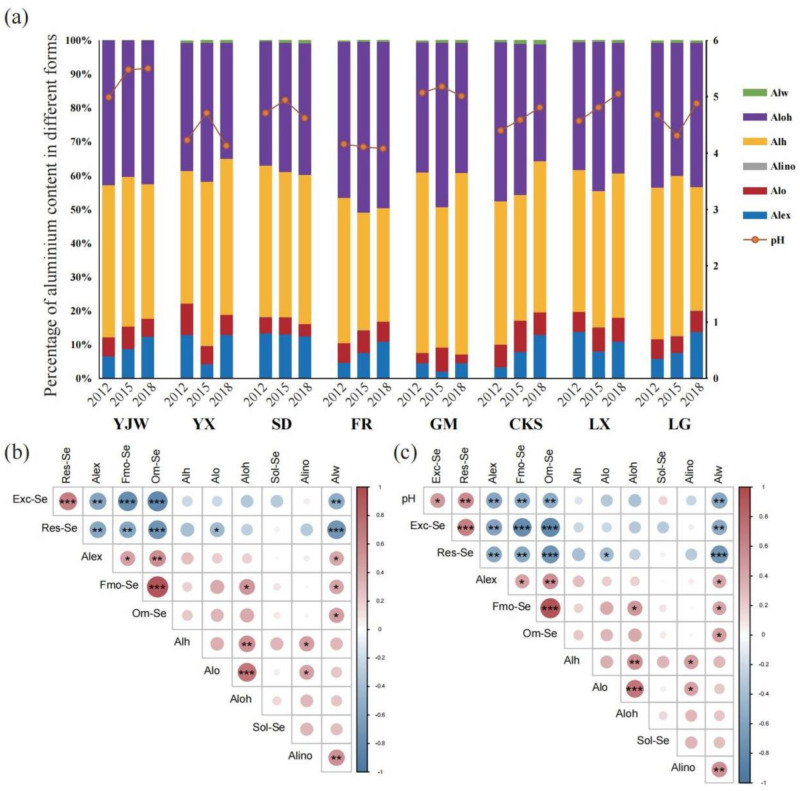
An illustration of the impact of soil pH fluctuations on Al morphology in tea plantations: (**a**) Proportions of Al morphology and pH values at eight sampling sites over three years; (**b**) Correlations between different forms of Se and Al; (**c**) Correlations among different forms of pH, Se, and Al. The symbols *, ** and *** indicate statistical significance at *p* < 0.05, *p* < 0.001 and *p* < 0.0001, respectively.

**Figure 3 plants-12-02882-f003:**
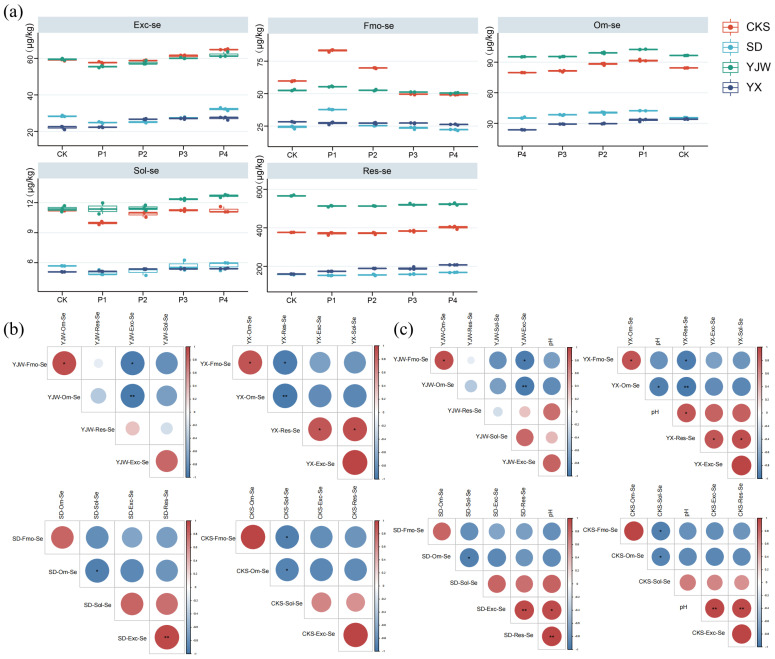
An illustration of the impact of pH fluctuations on Se morphology in response to simulated acid rain: (**a**) Variations in Se content across different forms within four soil types subjected to simulated acid rain; (**b**) correlation analysis between Se levels of various forms within four soil types exposed to simulated acid rain; (**c**) relationship between pH values of four soil types and Se levels across different forms under simulated acid rain. The symbols * and ** indicate statistical significance at *p* < 0.05 and *p* < 0.001, respectively.

**Figure 4 plants-12-02882-f004:**
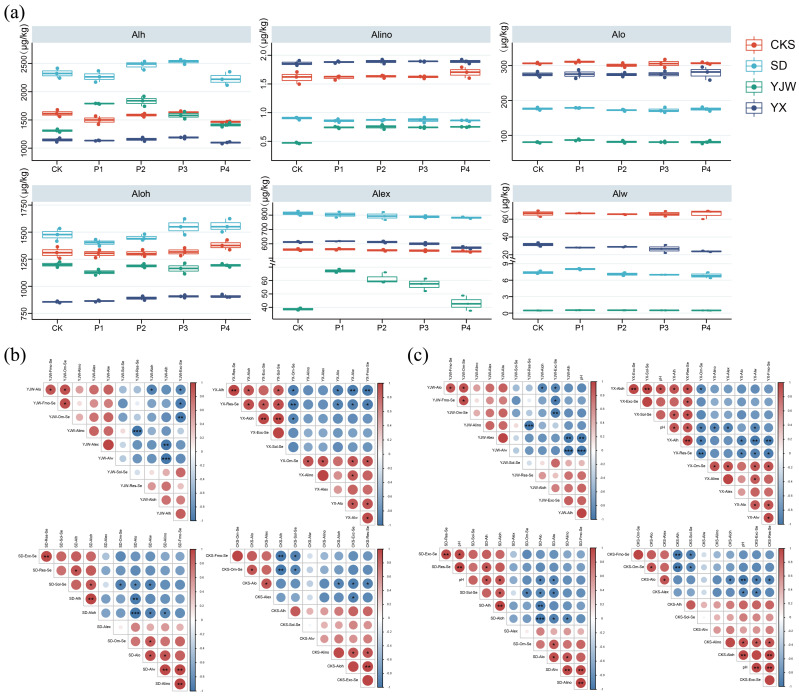
An illustration of the impact of pH variation on Al morphology in simulated acid rain conditions: (**a**) Alterations in the content of diverse forms of Al across four soils under simulated acid rain conditions; (**b**) association between different forms of Se and various forms of Al in four soils under simulated acid rain conditions; (**c**) relationship between pH levels and distinct forms of Se and Al across four soils under simulated acid rain conditions. The symbols *, ** and *** indicate statistical significance at *p* < 0.05, *p* < 0.001 and *p* < 0.0001, respectively.

**Figure 5 plants-12-02882-f005:**
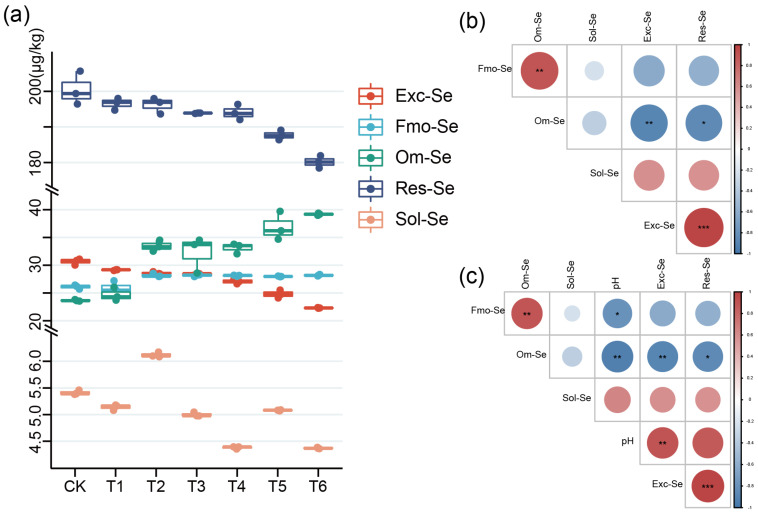
Effects of pH changes on Se morphology under different acidification methods: (**a**) Changes in Se content of different forms in soil under various acidification methods; (**b**) correlation between Se of different forms in soil under various acidification methods; (**c**) correlation between pH of soil and Se of different forms under various acidification methods. The symbols *, ** and *** indicate statistical significance at *p* < 0.05, *p* < 0.001 and *p* < 0.0001, respectively.

**Figure 6 plants-12-02882-f006:**
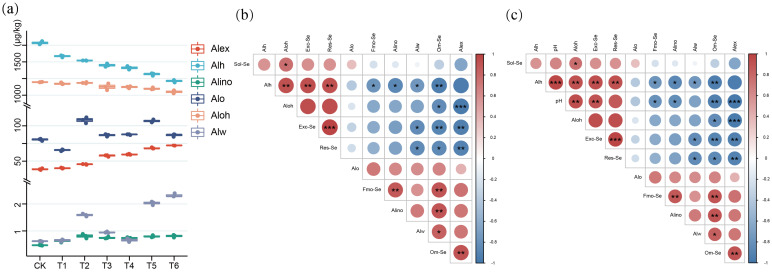
Effects of pH changes on Al morphology under different acidification modes: (**a**) Changes in the content of different forms of Al in soil under various acidification methods; (**b**) correlation between different forms of Al and different forms of Se in soil under various acidification methods; (**c**) correlation between pH in soil and different forms of Se and Al under various acidification methods. The symbols *, ** and *** indicate statistical significance at *p* < 0.05, *p* < 0.001 and *p* < 0.0001, respectively.

**Figure 7 plants-12-02882-f007:**
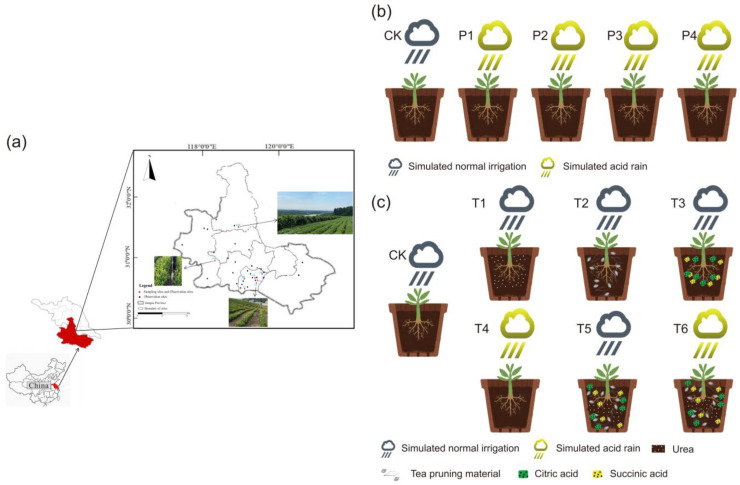
A schematic diagram of sampling points and experimental design for the tea plantation study. (**a**) Eight sampling points were selected for the field survey; (**b**) the simulated acid rain experiment was designed according to a specific schematic diagram; (**c**) different acidification methods were also tested using a specific experimental design.

## Data Availability

Not applicable.

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
