# Peer review of "Effect of Soil Acidification on the Production of Se-Rich Tea"

_plants, 2023, doi:10.3390/plants12152882_

Round 1

Reviewer 1 Report

The work presented by Yang et al. investigated the relationship between different forms of Se and Al in acidified soil of tea plantations. Results indicated that the total Se content was continually decreasing as the pH value declined. Exc-Se and Sol-Se are transformed into Fmo-Se and Om-Se, which is difficult to be absorbed by the tea plant. Furthermore, soil acidification could result in complexation effects of Alex and Exc-Se, thereby affecting the absorption of Se in the soil by tea plant. Generally, the study involves a fair amount of well-established experimental work from which the corresponding results are of importance for the related research. The observations in this study enhance our basic understanding the alteration of Se elements in the soil. However, I have some reservations on this manuscript.

Abstract should provide a clear summary consisting of specific objectives, methodology, results and interpretation in a descriptive manner. Furthermore, authors should present the significance and key findings of their study.

Some other mistakes:

Tea tree should be replaced with tea plant;

The format of the references is incorrect. The citation in the main text is also incorrect. 

Authors should revise all of those mistakes throughout this manuscript. 

Finally, authors should improve the English (language and gramma) to fit the standards of the journal.

Reviewer 2 Report

Journal: Plants (ISSN 2223-7747)

Manuscript ID: plants-2477998

Type: Article

Title: Effect of soil acidification on the production of selenium-rich tea

Authors: Bin Yang , Huan Zhang , Wenpei Ke , Jie Jiang , Yao Xiao , Jingjing Tian , Xujun Zhu , Lianggang Zong , Wanping Fang *

Section: Horticultural Science and Ornamental Plants

Special Issue: Tea Plants Cultivation

Selenium (Se) is an essential trace element for humans and animals, and it plays an important role in immune regulation and disease prevention. Tea is one of the top three beverages in the world, and it contains active ingredients such as polyphenols, theanine, flavonoids, and volatile substances, which have important health benefits. The tea tree has suitable Se aggregation ability, which can absorb inorganic Se and transform it into safe and effective organic Se through absorption by the human body, thereby improving human immunity and preventing the occurrence of many diseases. 

The above topic of research work is investigated in the literature, and there is a very few of reference published. However, this paper gives significant contribution to the current knowledge in related field. The data are sound and it deserves to be published, after major revisions.

I recommend considering the manuscript for publication upon addressing following major observations:

Abstract

Q1. Page-1, Line-19: Use the passive voice not possessive (our)??

Q2. Statistical design is not mentioned here...found missing??

Introduction

Q3. Page-1, Line-36: Don’t start a sentence with an abbreviation here??? Write Selenium instead of Se.

Q4. The authors are completely failed to build the hypothesis??? Why the present research is planned??

Materials and methods

Q5 .The text has many typing and grammatical errors, capitalization issues.

Q6. All proper nouns must be abbreviated. Abbreviations must be described completely at first mention with brackets.

Q7. Quality assurance of data is mandatory!!! How many batch, repeats, chemical grade and for used instruments manufacturers’ user manual and instructions were strictly followed or not!!!

Q8. The methodology section warrants concise description followed by only one or two names of the reference manuals used for the analyses. Statistical methods should be briefly described and does not mingle with other technical obtained data.

Results and discussion

Obtained data is sound and must be published.

Q9. The results and discussion should be written succinctly with support from relevant references only. Tea is the test crop in this study and the references used should be predominantly related to tea.

Q10. I would have expected slightly greater discussion of how exactly tea growth was affected; more detail on the mechanisms and logical reasoning is required??

Q11. For discussion section, not much detailed discussion is going on. This is just restating the observations and results. There is much more scope here for discussing the implications of what these results mean??

Conclusion

Q12. Novelty of this research work is again questionable??

Q13. It seems a few speculations in the end of paragraph??

References

Q14. A few very old references have been used??? These must be updated with recent research findings or removed. Proper formatting is questionable??? It must be according to MDPI Plants Journal.

Q15. References formatting are inconsistent. A few DOI missing?? Verify each reference from original source and cross check references in the text and reference section.

English style and language requires a profound revision. However, the readability of the manuscript needs to be improved, preferably carefully reviewing by a native English speaker. 

Reviewer 3 Report

The paper „Effect of soil acidification on the production of selenium-rich tea” is current and very well structurate.

The authors highlighted very well the impact of Al morphology on Se morphology during soil acidification. Also, they highlighted very well the relationship between Se and Al morphology with respect to soils pH in tea gardens.

I propose the publish of the paper after they will expanding the conclusions. In the Conclusion, briefly add the science of your research, benefits, advantages-disadvantages, practical application. Perspectives in the continuation of research and trials.

Round 2

Reviewer 2 Report

Suggestions have been incorporated.

Minor editing of English language required
